



# Particle Clustering and Subclustering as a Proxy for Mixing in Geophysical Flows

Rishiraj Chakraborty[1], Aaron Coutino[1], and Marek Stastna[1]

[1]Department of Applied Mathematics, University of Waterloo

**Correspondence:** Rishiraj Chakraborty (r25chakr@uwaterloo.ca)

**Abstract.** The Eulerian point of view is the traditional theoretical and numerical tool to describe fluid mechanics. Some modern computational fluid dynamics codes allow for the efficient simulation of particles, in turn facilitating a Lagrangian description of the flow. The existence and persistence of Lagrangian coherent structures in fluid flow has been a topic of considerable study. Here we focus on the ability of Lagrangian methods to characterize mixing in geophysical flows. We study the instability of

a strongly non-linear double jet flow, initially in geostrophic balance, which forms quasi-coherent vortices when subjected to ageostrophic perturbations. Particle clustering techniques are applied to study the behaviour of the particles in the vicinity of coherent vortices. Changes in inter–particle distance play a key role in establishing the patterns in particle trajectories. This paper exploits graph theory in finding particle clusters and regions of dense interactions (also known as sub-clusters). The methods discussed and results presented in this paper can be used to identify mixing in a flow and extract information about

particle behaviour in coherent structures from a Lagrangian point of view.

## 1   Introduction

There are two different geometric approaches to fluid mechanics, the Eulerian and the Lagrangian approach. In the Eulerian approach, field values are obtained on a spatial grid, for example from numerical simulation output. In the Lagrangian approach measurement data is obtained following the fluid, as in the case of temperature measurements by a weather balloon. Many

naturally occurring flows are complex, three–dimensional and at least to some extent, turbulent. Such flows are characterized by a richness of vorticity and the rapid mixing of passive tracers as discussed in (Davidson, 2015), chapter 3. At the same time, satellite imagery suggests large scale flows exhibit prominent coherent patterns, and this is theoretically supported by the so-called inverse cascade of two dimensional turbulence in which energy moves to larger scales while enstrophy moves to smaller scales (Davidson, 2015), chapter 10.

Even three dimensional turbulent flows are known to contain quasi-deterministic coherent structures (Hussain, 1983). Coherent structures can be thought of as turbulent fluid masses having temporal correlation in vorticity over some spatial extent (e.g. a shear layer in a flow). **Fig.**(1) shows the evolution of the enstrophy field of a two dimensional double jet initially in geostrophic balance, subjected to ageostrophic perturbations. The evolution depicts the formation of vortices due to instability of the geostrophic flow. Coherent structures like vortices and filaments, undergo frequent stretching and folding. The identi-

fication of coherent structures in turbulent flows gave the revolutionary notion in fluid mechanics that turbulent flows are not



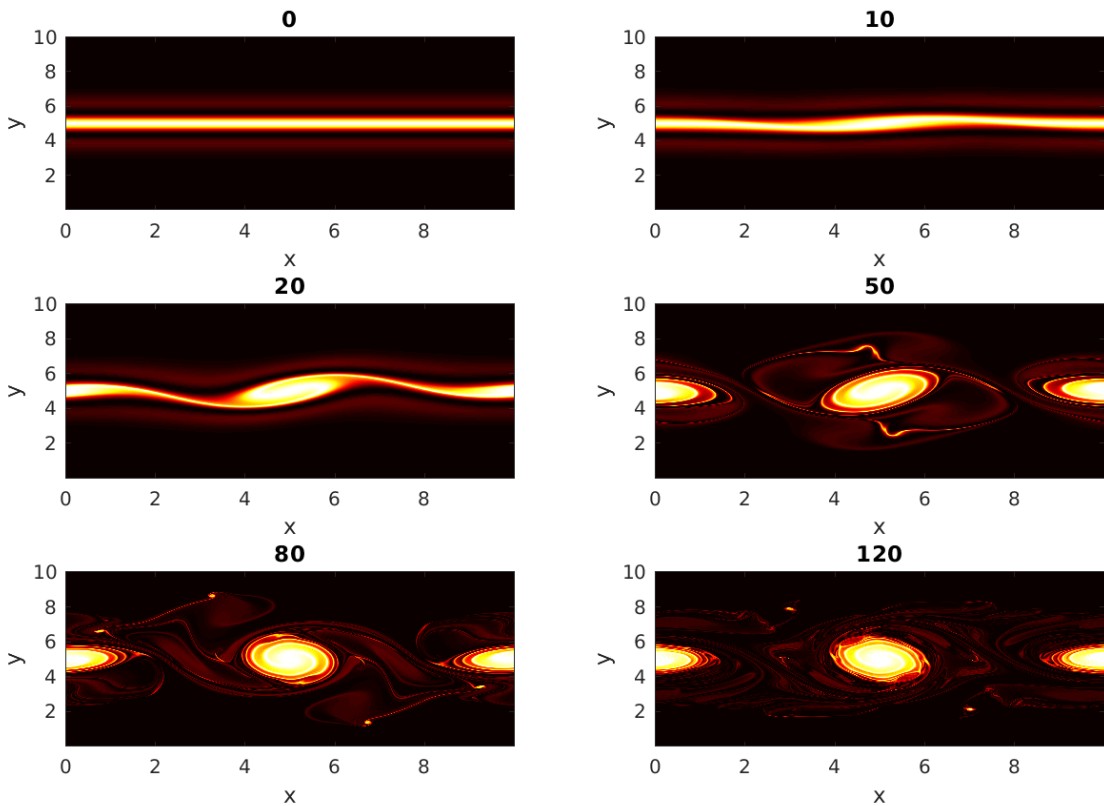

**Figure 1.** The enstrophy field showing the evolution of the unstable double jet with time. The bright areas indicate regions of high enstrophy which are found between the two jets at early times.

completely random but can contain orderly organized structures and these coherent structures in specific regions can influence mixing, transport and other physically relevant features (Kline et al., 1967).

The study of coherent flow structures has received significant interest in the recent past. The existing methods for detecting coherent behaviour mathematically are either geometric or probabilistic; (Allshouse and Peacock, 2015) discusses and compares the different methods. Geometric methods aim to find distinct boundaries between the coherent structures, whereas probabilistic methods use the concept of sets with minimal dispersion moving in a flow to identify coherent structures. (Padberg-Gehle and Schneide, 2017) in their Introduction, however, note that existing methods for finding coherent structures require the full knowledge of the flow-field and the underlying dynamical system. This, in turn, requires high resolution trajectory data. This can be numerically expensive, as well as challenging to find in applications. (Hadjighasem et al., 2017), in their review of various Lagrangian techniques for finding coherent structures, say that the Lagrangian diagnostic scalar field methods are incapable of providing a strict definition of coherent flow structures and are also not effective in establishing a precise





mathematical connection between the geometric features and the flow structures. Such diagnostic methods include: Finite time Lyapunov exponents (FTLE), Finite-Size Lyapunov Exponent (FSLE), Mesochronic analysis, Trajectory length, Trajectory complexity and Shape coherence. (Hadjighasem et al., 2017) also describes the various methods of applying mathematical coherence principles to locate coherent structures. However, these principles only apply in the early stages of the flow evolution,

it is not guaranteed that the coherence principles comply with observed coherent patterns at later times. Examples of mathematical coherence principles include transfer operator methods like the probabilistic transfer operator and the dynamic Laplace operator. These methods identify maximally coherent or minimally dispersive (not dispersive in the sense of wave theory) regions over a finite time interval. Such regions are expected to minimally mix with the surrounding phase space and are named "almost-invariant sets" for autonomous systems and "coherent sets" for non-autonomous systems. A different mathematical

approach is the hierarchical coherent pairs method, which initially splits a given domain into a pair of coherent sets using the transfer operator method, and then subsequently refines the coherent sets iteratively. This is accomplished using the probabilistic transfer operator. The iteration is carried out until a reference measure of the probability, $\mu$, falls below a user defined cut-off. A third category of mathematical approaches for finding coherent structures based on Lagrangian data is clustering. (Hadjighasem et al., 2017) reviews the Fuzzy C-means clustering of trajectories by (Froyland and Padberg-Gehle, 2015) which

uses the traditional fuzzy C-means clustering to identify finite-time coherent structures and mixing in a flow. This method uses trajectories of Lagrangian particles, over discrete time-intervals, and applies the Fuzzy C-means algorithm to locate coherent sets as clusters of trajectories according to the dynamic distances between trajectories. Another similar method for locating coherent structures is the spectral clustering of trajectories as proposed by (Hadjighasem et al., 2016) and implemented by (Padberg-Gehle and Schneide, 2017). (Mancho et al., 2004) discusses algorithms to compute hyperbolic trajectories from data

sets on oceanographic flows and how to locate their stable and unstable manifolds. (Mendoza and Mancho, 2010) also discusses how phase portraits obtained using Lagrangian descriptors can provide a representation of the interconnected features of the underlying dynamical system. (Rose et al., 2015) uses a coupled implementation of a mix of Eulerian and Lagrangian models for simulating the full life cycles of fish species anchovy and sardine in the California Current Systems. The Lagrangian model used is an individual fish based model which tracks each fish of every species. (Padberg-Gehle and Schneide, 2017) used a

generalized graph Laplacian eigenvalue problem to extract coherent sets from several fabricated examples (e.g. Bickley jet) as well as measured data. The authors also highlighted regions of strong mixing in flow, using local network measures like node degree and the local clustering coefficient. These local network measures provide information for each Lagrangian particle. Inspired by these, we wish to extract regions of dense mixing in flow using a graph theoretic network approach and compare the results with those obtained from spectral clustering. We also wish to use an evolving simulation for which coherent regions

evolve dynamically through stretching and folding and are not known *a priori*.

From an Eulerian point of view, mixing can be characterized by studying the advection-diffusion equation for a passive tracer $\theta$ (Salmon, 1998),

$$\frac{\partial \theta}{\partial t} + v \cdot \nabla \theta = \kappa \nabla^2 \theta \tag{1}$$





where $v$ is the fluid velocity and $\kappa$ is the diffusion coefficient. Mixing and stirring depends on the gradient of $\theta$ and the hence the extent of mixing and stirring in a given domain for a given flow can be measured by the spatial variability index

$$C = \frac{1}{2} \int \int \nabla\theta.\nabla\theta dx. \tag{2}$$

Taking the time derivative of $C$, and following the simplification procedure in (Salmon, 1998), we obtain,

$$\frac{dC}{dt} = \int \int [(v.\nabla\theta)\nabla^2\theta - \kappa(\nabla^2\theta)^2] dx \tag{3}$$

Fundamentally, mixing is a result of molecular diffusion, and hence the diffusive (second) term in equation 3 represents the effect of mixing, while the first term containing the gradient of $\theta$ represents the effect of stirring. This implies that an initial high value of $\nabla\theta$ will promote mixing and hence diffusion, which in turn will to lead to a decrease in $\nabla\theta$. This can also be verified from a dynamical systems point of view. (Prants, 2014) in his review paper describes mixing as follows. Let us consider the basin $A$ with a circulation where there is a domain $B$ with a dye occupying at t = 0 the volume $V(B_0)$. Let us consider a domain $C$ in A. The volume of the dye in the domain $C$ at time $t$ is $V(Bt \cap C)$, and its concentration in $C$ is given by the ratio $V(B_t \cap C)/V(C)$. Full mixing is defined in the sense that in the course of time, for any domain $C \in A$, the concentration of the dye is the same as in every other region in $A$. However, calculating the true three-dimensional Eulerian flow field, and the distribution of $\theta$, for an actual geophysical flow (e.g. a hurricane) is an impossible task. This is due to the immense range of scales that typifies naturally occurring fluid motions. If one considers a hurricane, active scales range from hundreds of kilometers to sub millimeter scales. Many models in geophysical fluid dynamics thus focus on representing the coherent scales of motion. In such cases the fundamentally three dimensional motions that would carry out efficient mixing are filtered out during the theoretical derivation of the governing equations. A Lagrangian approach to mixing, based on particle proximity, may thus be more profitable. This is because it allows for an idealized representation of the three-dimensional turbulence that is ignored by the governing equations .

(Klimenko, 2009) provides an example of this approach to describe mixing. His idea is stochastic, where each particle has a deterministic component of motion governed by the known flow field and a random walk component. The particles are assigned scalar properties which can change due to mixing. The random walk component depends on the joint probability distribution of the particle as functions of position and the scalar properties. In his equation (36) the author defines the intensity of mixing between two particles as proportional to the distance between the particles in physical space. Inspired by (Klimenko, 2009), we use a numerically inexpensive version of this idea, by loosely saying that, there is some non-zero probability of mixing with exchange of properties taking place between two particles that approach below a given threshold and a qualitative measure of mixing is given by interaction among particles. Interaction once occurred, is counted as a unit of mixing and our hypothesis says that, if we have three particles, say, $A$, $B$ and $C$, and if particle $A$ interacts with particle $B$ and if particle $B$ interacts with particle $C$, then indirectly, particle $A$ has interacted with particle $C$, to some extent. We then extend this idea to the assumption that a region comprising of a higher number of interacting particles corresponds to one with higher probabilities of mixing. The technical details are discussed in section [2.3].

The remaining parts of the paper are structured in the following manner. Section [2] discuses the methods used in our work including the governing equations and description of the numerical code used to solve them. This is followed by the methods



for clustering particles (section 2.2), identifying regions of mixing (section 2.3) and the methods for spectral clustering (section 3.4). Section [3] presents a detailed discussion of the results obtained by implementing each of the methods above and also draws relevant comparisons as needed. The final section [4] concludes the work and highlights the major findings.

## 2 Methods

### 5  2.1 Governing Equations and Numerical Methods

We consider the shallow water equations on the f–plane (Kundu et al. (2008)). All simulations are carried out with a code developed in house using CUDA, called CUDA Shallow Water and Particles (cuSWAP), which provides numerical solutions to the Shallow Water equations. CUDA is a C/C++ based parallel computing platform developed by NVIDIA to harness the computational power of GPUs (Nickolls et al., 2008). We choose to solve these equations using spectral methods to take

advantage of the cuFFT library (Nvidia, 2010). This code solves the governing equations in a doubly periodic domain with variable topography. The IO is handled using NETCDF. The time-stepping scheme is a low-memory Huen's Method (Ascher and Petzold, 1998). This code also has a Lagrangian attribute which performs particle tracking using cubic interpolation and symplectic Euler time-stepping (Mickens, 2000). Additionally this code dynamically calculates and outputs neighbours of a particle based on inter-particle distance. This data represents particle interactions and is used to construct adjacency matrices

relevant to our work as described in section [2.2].

The shallow water equations, written out in the form amenable to numerical solution with an FFT-based method, express the conservation of mass

$$\frac{\partial \eta}{\partial t} + (H + \eta)\left(\frac{\partial u}{\partial x} + \frac{\partial v}{\partial y}\right) + u\left(\frac{\partial H}{\partial x} + \frac{\partial \eta}{\partial x}\right) + v\left(\frac{\partial H}{\partial y} + \frac{\partial \eta}{\partial y}\right) = 0,$$

and the conservation of linear momentum,

$$\frac{\partial u}{\partial t} + u\frac{\partial u}{\partial x} + v\frac{\partial u}{\partial y} - fv = -g\frac{\partial \eta}{\partial x},$$

$$\frac{\partial v}{\partial t} + u\frac{\partial v}{\partial x} + v\frac{\partial v}{\partial y} + fu = -g\frac{\partial \eta}{\partial y},$$

where $\eta(x,y,t)$ is the perturbation height field, $H(x,y)$ is the bottom topography (taken as constant throughout the present work), $(u(x,y,t), v(x,y,t))$ are the velocity field, $f$ is the rotation rate taken as constant (i.e. the f–plane), and $g$ is the acceleration due to gravity. The pressure field is hydrostatic.

The initial conditions consist of a geostrophically balanced jet and an ageostrophic perturbation with a radially symmetric form. The exact functional form of the perturbation was not found to be important for triggering the instability of the jet. The functional form of the initial conditions is given by,





$$u(x,y,0) = 2ga_0 \frac{\tanh(y)}{\cosh^2(y)}$$

$$v(x,y,0) = 0$$

$$\eta(x,y,0) = a_0 \left( \frac{1}{\cosh^2(y)} + \frac{1}{\cosh^8(\sqrt{x^2+y^2}/2)} \right)$$

where $a_0 = 0.1H_0$. The two relevant dimensionless numbers are the Froude number and Rossby number,

$$5 \quad Fr = \frac{U}{\sqrt{gH}} \approx 0.17,$$

$$Ro = \frac{U}{fL} \approx 0.3775,$$

Results will be reported in dimensionless form. The simulation is thus carried out in a square domain with side dimension 10. The resolution used is $2048 \times 2048$ and the number of particles tracked is $400 \times 400$, initially distributed uniformly in a grid pattern. The resolution is fine enough to represent both the primary, vortex generating instability, and the filaments formed 10  from the interaction between vortices.

## 2.2 Clustering particles

Clustering the particles in a flow means we group the particles based on some form of particle behaviour we wish to identify. In this paper we target the phenomenon of mixing in a flow by measuring instances of particle-particle proximity below a threshold. The inter-particle interactions we employ fall under the category of binary classification, i.e. two particles have 15  either interacted or they have not. We set a threshold inter-particle distance $\epsilon$ such that at some given time, if the distance between any two particles becomes less than $\epsilon$, those two particles will be said to have interacted with each other at that time. For mixing, it is natural to demand that the value of $\epsilon$ is less than grid spacing (though note that (Padberg-Gehle and Schneide, 2017) in fact demand $\epsilon$ to be greater than the grid spacing for spectral clustering). Thus, for every time step we search for particles which are within a radial distance of $\epsilon$ from every particle. A natural mathematical way to represent this 20  information is to build a matrix. These matrices are known as adjacency matrices which are symmetric square matrices with dimensions *(number of particles$^2$)*. Each row in an adjacency matrix corresponds to a particle and the columns correspond to all the particles this particle may interact with. If particle 'i' is said to have interacted with particle 'j', then the adjacency matrix, an initially zero matrix, will have 1 in cells $(i,j)$ and $(j,i)$. **Fig.** (2) demonstrates a tutorial example of how to construct an adjacency matrix from particle interactions. There are two ways in which we create an adjacency matrix in our work:

25  – *Cumulative adjacency matrix:* One interaction between two particles in the entire time span will yield a permanent 1 in the corresponding cells of the particles in the matrix.

– *Instantaneous adjacency matrix:* One interaction between two particles at a particular time will yield a temporary 1 in the corresponding cells of the particles in the matrix. This type of matrix is refreshed every output time and new 1s and 0s are registered for the new output time.



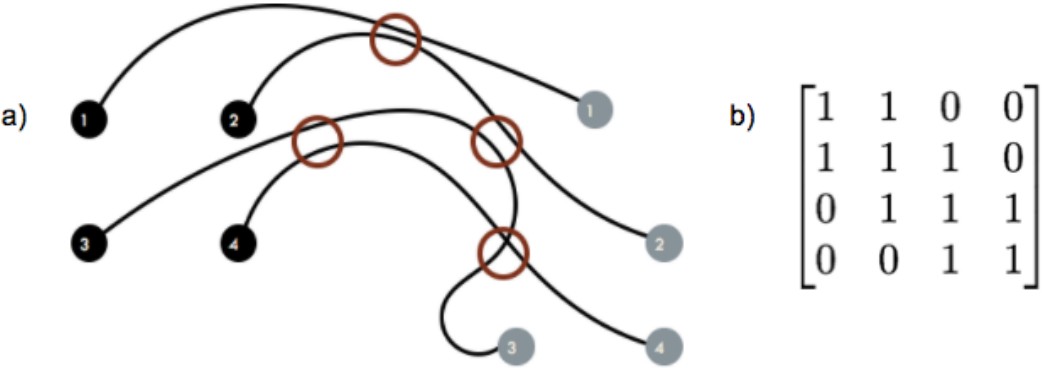

**Figure 2.** a) Idealized Lagrangian paths of four particles showing where they have interacted along the course of their paths. b) Adjacency matrix corresponding to the particle interactions shown in part (a)

Before we describe how we cluster these particles based on their interactions, we quickly introduce graphs from discrete mathematics. A graph is a structure which has a set of objects and some objects may be related to each other in some way. The objects are called nodes, and if two nodes are related to each other in some way, they are connected by an edge. Mathematically, a graph is represented in the form of an ordered pair $G = (V, E)$ where $V$ is a set of vertices or nodes and $E$ is set of edges

which consists of two element subsets of $V$. An adjacency matrix can be converted into a graph with the particles forming the nodes and the interactions forming the edges. Looking at **Fig.**2, we construct a corresponding graph shown in **Fig.**3

A graph formed from an adjacency matrix of particle interactions, can be used to cluster the particles by finding connected components in a graph. To demonstrate this concept, we add two more nodes to the graph in **Fig.**3. The way they are added is shown in **Fig.**4. It is seen that the graph can be visually split into two parts as marked by the ellipses. These are two separate,

connected components in our imaginary graph. The connected components in a graph can be mined by using a standard depth first search algorithm. We carry out this procedure on the graph in our problem using MATLAB . The different connected components in the graph form the different clusters. In regards to our earlier point of mixing we see that each cluster has particles that have interacted with at least another particle inside the cluster and thus odds are high that some mixing may be happening among particles within these clusters. This gives us a level one classification of particles which will later help us

track down regions of mixing.





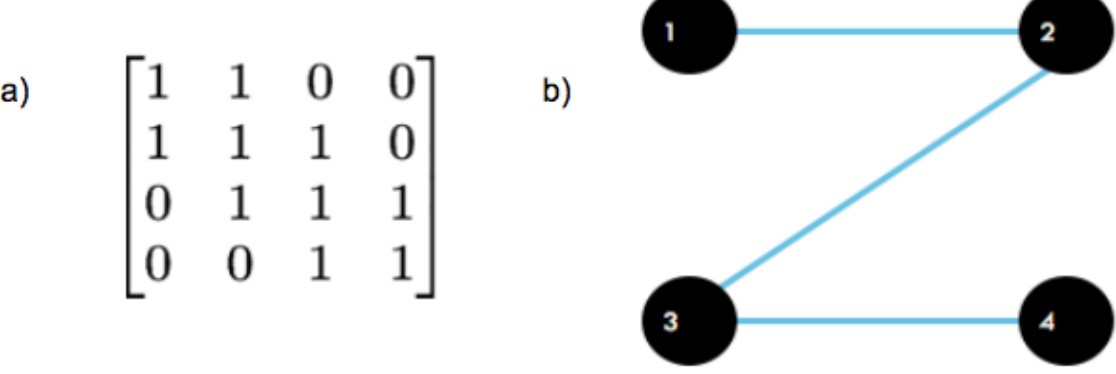

**Figure 3.** a) Adjacency matrix of four particles. b) Graph corresponding to part (a)

## 2.3 Mining dense sub-clusters from a cluster

Until this point, clusters have been based on inter-particle interactions. Though, these clusters tell us about which particles interacted, they do not tell us anything about the degree or intensity of interaction. We want to find regions in the flow where there are higher intensities of mutual interactions among particles compared to rest of the flow. We consider a cumulative
cluster, which is a connected graph and use the pruning algorithm *Quick* described by (Liu and Wong, 2008) to look for dense sub-clusters within this cluster.

A clique is a graph whose nodes are all connected to each other, hence a clique is $100\%$ dense. The minimum degree of a graph is the minimum number of neighbors that a node has in the graph. Let the minimum degree be denoted by $deg_{min}$ and $N$ be the size of the graph. A $\gamma$-quasi clique is a graph which satisfies:

$$deg_{min} >= \gamma[N-1] \tag{4}$$

The density of a sub-graph is based on the following parameters:

- The density parameter $\gamma$, such that (4) is satisfied.

- Minimum size of a subgraph. The algorithm will only look for solutions whose sizes are greater than or equal to the specified minimum size parameter, $min\_size$.

All subgraphs mined, hence, have a minimum degree greater than or equal to $\gamma(min\_size - 1)$. These two parameters drive how many minimum particles we want from a dense sub-cluster to have interacted with a particle in the same dense sub-cluster.

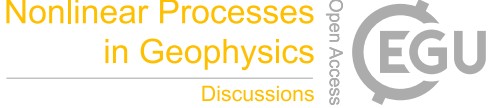



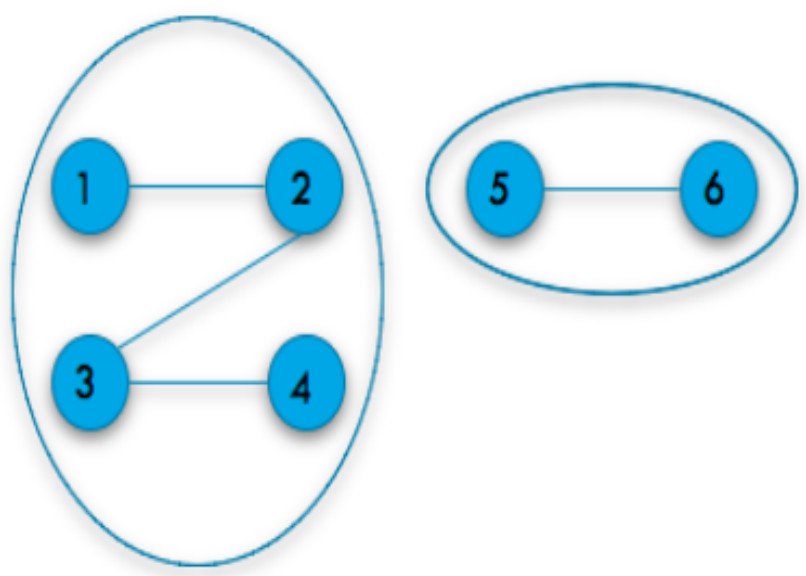

**Figure 4.** Graph split into its connected components.

We find sub-clusters with a minimum size of 20 and $\gamma = 0.25$, so that the minimum degree is at-least 5. There are cases where subsets of a bigger $\gamma$-quasi clique are also $\gamma$-quasi cliques. The algorithm *Quick* makes sure that it mines only the maximal $\gamma$-quasi cliques for a specified $\gamma$. The algorithm is described in the next subsection.

Fig. (5) shows an example of how dense sub-clusters are mined. The connected graph in **Fig.** (5) can be a considered as

5   a small illustration of an actual cumulative cluster of particles. For an arbitrary $\gamma = 0.3$ and minimum size of the sub-graphs equal to 3, the algorithm shows that the nodes inside the dotted black circles are dense sub-graphs inside the graph. In the context of Lagrangian fluid mechanics, interactions among particles in these sub-clusters are much denser than other regions in the flow.

### 2.3.1 Description of the Quick algorithm

10   We will now introduce graph theoretic terminology that we will be required in the following section. This work is based on (Liu and Wong, 2008).

A *graph* $G$ is an ordered pair of sets $(V, E)$, where $V$ is a set of vertices and $E$ is a set of edges joining the vertices.

*Neighbours* of a vertex $v$ in $G$ are denoted by $N^G(v)$ which are the nodes adjacent to $v$ in $G$.

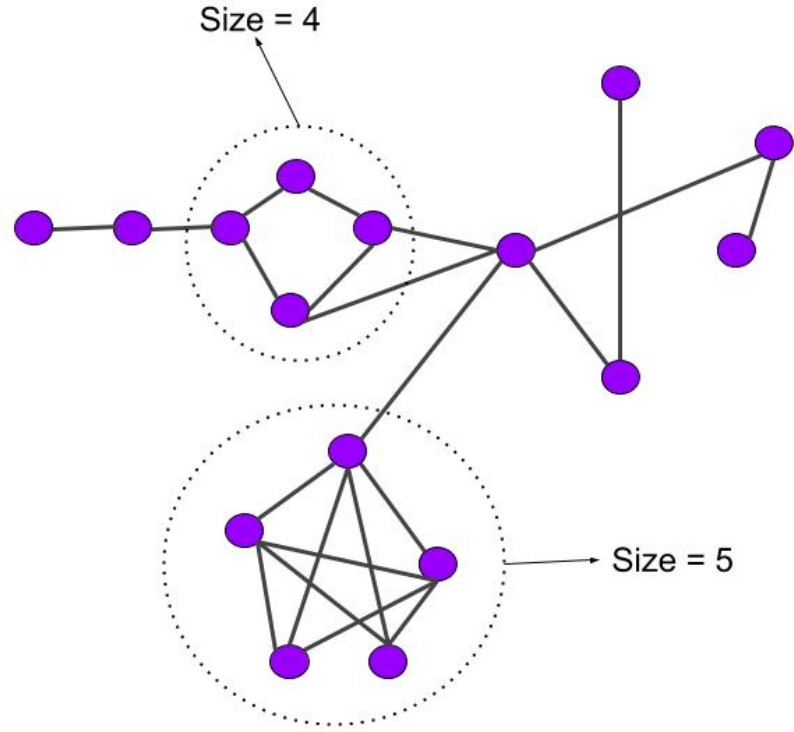

**Figure 5.** A connected graph symbolizing a scaled down version of a cumulative cluster; The black dotted circles denote the dense-sub graphs for an arbitrary $min\_size = 3$ and $\gamma = 0.3$

The *degree* of a vertex $v$ in $G$, denoted by $deg^G(v)$, is the number of neighbours of $v$, $|N^G(v)|$.

The *distance* between two vertices $u$ and $v$ in $G$, denoted by $dist^G(u,v)$, is the number of edges on the shortest path from $u$ to $v$.

For a vertex $v$ in $V$, $N_k^G(v) = \{u | dist^G(u,v) \le k\}$ denote the $k$-nearest neighbours of $v$.

5     The *diameter* of a graph $G$, denoted by $diam(G)$ is defined as $max_{u,v \in V} dist^G(u,v)$.

For any vertex set $\{X | X \subset V\}$, $cand\_exts(X)$ represents the set which contains vertices that can be used to extend the set $X$ in order to form a $\gamma$-quasi clique.



For a vertex $u$ in a vertex set $X$, $indeg^X(u)$ represents the number of neighbours of $u$ in $X$ and $exdeg^X(u)$ represents the number of neighbours of $u$ in the set $cand\_exts(X)$.

The minimal degree of vertices in $X$, denoted by $deg_{min}(X)$, is $min\{indeg^X(v) + exdeg^X(v)|v \in X\}$.

It follows from the definition of a $\gamma$-quasi clique that the maximal number of vertices in $cand\_exts(X)$ that can be added

to $X$ concurrently, is less than $U_X^{min} = \lfloor deg_{min}(X)/\gamma \rfloor + 1 - |X|$.

In another case where, vertex $u \in X$ and $indeg^X(u) < \lceil \gamma(|X|-1) \rceil$, it becomes apparent that at-least some vertices must be added to $X$ so it can be extended to form a $\gamma$-quasi clique. This lower bound is denoted by $L_X^{min}$. Let $indeg_{min}(X) = min\{indeg^X(v)|v \in X\}$, then $L_{min}^X$ is defined as $min\{t|indeg_{min}(X) + t \geq \lceil \gamma(|X|+t-1) \rceil$

*Quick* uses several effective pruning techniques to eliminate vertices from $cand\_exts(X)$ of a vertex set $X$. Valid extensions

are added to $X$, to check if the new vertex set $(X \cup cand\_exts(X))$ satisfies the $\gamma$-quasi clique criterion. The following pruning techniques form an essential part of *Quick* algorithm. The proof of the Lemmas used by these techniques can be found in (Liu and Wong, 2008).

✳ Depending on $\gamma$, we find a $k$ such that vertices not in $\bigcap_{v \in X} N_k^G(v)$ are removed from $cand\_exts(X)$. This is called pruning based on diameter.

✳ We use the Cocain algorithm (Zeng et al., 2006) to eliminate all such vertices $u$ from $cand\_exts(X)$ who satisfy $indeg^X(u) + exdeg^X(u) < \lceil \gamma(|X| + exdeg^X(u)) \rceil$. This is because, neither such a vertex $u$ nor any of its neighbours in $cand\_exts(X)$, if added, will satisfy the $\gamma$-quasi clique criterion.

✳ We set an upper bound $U_x$ based on $U_X^{min}$, such that, $U_X = max\{t| \sum_{v \in X} indeg^X(v) + \sum_{1 \leq i \leq t}\}indeg^X(v_i) \geq |X|\lceil \gamma(|X| + t - 1) \rceil, 1 \leq t \leq U_X^{min}\}$, where $v_i$ are vertices in $cand\_exts(X)$ sorted in descending order of their $indeg^X$ value.

If vertex $u \in cand\_exts(X)$ and $indeg^X(u) + U_X - 1 < \lceil \gamma(|X| + U_X - 1) \rceil$, such a vertex $u$ can be pruned from $cand\_exts(X)$. Otherwise, if $u \in X$ and $indeg^X(u) + U_X < \lceil \gamma(|X| + U_X - 1) \rceil$, then $\gamma$-quasi cliques cannot be generated by extending $X$.

✳ We set a lower bound $L_X$ based on $L_X^{min}$, such that, $L_X = min\{t| \sum_{v \in X} indeg^X(v) + \sum_{1 \leq i \leq t}\}indeg^X(v_i) \geq |X|\lceil \gamma(|X| + t - 1) \rceil, L_X^{min} \leq t \leq n\}$, if such $t$ exists, else $L_x = |cand\_exts(X)| + 1$. If vertex $u \in cand\_exts(X)$ and $indeg^X(u) +$

$exdeg^X(u) < \lceil \gamma(|X|+L_X-1) \rceil$, such a vertex $u$ can be pruned from $cand\_exts(X)$. Otherwise, if $u \in X$ and $indeg^X(u) + exdeg^X(u) < \lceil \gamma(|X|+L_X-1) \rceil$, then $\gamma$-quasi cliques cannot be generated by extending $X$. Before performing the above checks, we also check if $L_X > U_X$, and if true there is no need to extend $X$ further.

✳ In a vertex set $X$, if we have a vertex $v \in X$ such that $indeg^X(v) + exdeg^X(v) = \lceil \gamma(|X| + L_X - 1) \rceil$, then $v$ is called a critical vertex of $X$. If $G(Y)$ is a $\gamma$-quasi-clique and $v$ is a critical vertex, we have $\{u|(u,v) \in E \wedge u \in cand\_exts(X)\} \subseteq$

$Y$. Hence, whenever we encounter a critical vertex in our vertex set $X$, we instantly add it's neighbours present in $cand\_exts(X)$ to $X$.





✷ We are mining exclusively maximal $\gamma$-quasi-cliques and it can be proved that if $u$ is a vertex in $cand\_exts(X)$ such that $indeg^X(u) \geq \lceil \gamma|X| \rceil$ and if for any $v \in X$ such that $(u,v) \notin E$, we have $indeg^X(v) \geq \lceil \gamma|X| \rceil$, then for any vertex set $Y$ such that $G(Y)$ is a $\gamma$-quasi-clique and $Y \subseteq (X \cup (cand\_exts(X) \cap N^G(u) \cap (\bigcap_{v \in X \wedge (u,v) \notin E} N^G(v))))$, G(Y) cannot be a maximal $\gamma$-quasi-clique. So we use $C_X(u) = (cand\_exts(X) \cap N^G(u) \cap (\bigcap_{v \in X \wedge (u,v) \notin E} N^G(v)))$ to denote the vertices covered by $u$ and $u$ is called the cover vertex of $X$. We find $u$ such that it maximizes $C_X(u)$, put the vertices in $C_X(u)$ at the end of $cand\_exts(X)$ and then use the vertices in $cand\_exts(X) - C_X(u)$ to extend X.

## 2.4 Spectral Clustering

Here we explore a different method of sub-clustering a cumulative cluster that does not require the threshold spacing $\epsilon$ to be greater than the grid spacing. Once we identify a cumulative cluster, we extract the portion of the adjacency matrix corresponding to particles exclusively within it. Let's suppose we name this adjacency matrix $A$. We find the degree matrix, $D$ which is a diagonal matrix with $D_{ii} = d_i$, where $d_i$ is the degree of the node $x_i$, i.e., $D_{ii} = \sum_{j=1}^{n} A_{ij}$, the number of neighbours of node $i$. The non-normalized graph Laplacian is given by $L = D - A$, and the normalized graph Laplacian is given by $\mathcal{L} = I_n - D^{-\frac{1}{2}} A D^{-\frac{1}{2}}$. The eigenvalues of $\mathcal{L}$ are real and non-negative and are in the order $0 = \lambda_1 \leq \lambda_2 \leq \lambda_3 \leq ... \leq \lambda_n$. The second smallest eigenvalue $\lambda_2$ is called the algebraic connectivity (Fiedler, 1973) of a graph and can only be non-zero if the graph is connected. We expect that to be true in our case as the cumulative cluster corresponds to a connected graph. Spectral clustering is expected to help find coherent structures in fluid transport, which in lay-man's terms mean particles whose trajectories stay close to each other or interact more often. The mathematics in this section is the outcome of solving a balanced cut problem in a network (Hadjighasem et al., 2016). So the idea is if $\lambda_2$ is the only eigenvalue close to zero then the graph is nearly decoupled into two communities. Similarly if all $\lambda_i$, $i = 2, 3, ...k$ for some $k < n$ are close to zero and there is a spectral gap between $\lambda_k$ and $\lambda_{k+1}$, then the cluster is nearly separated into $k$ communities. The corresponding eigenvectors carry information about the division of these particles. Hence, we capture these eigenvectors, performing a dimensional reduction on our data, and apply unsupervised clustering on them. We employ the standard *k-means clustering algorithm* (Lloyd, 1982) on the reduced data to identify the different communities. Since we are already in a cumulative cluster, and the further clustering is supposed to reveal the coherent structures in the flow, we expect to find the regions with a comparatively higher intensity of interaction. However, since we use *k-means clustering*, we do not expect it to identify precise locations of solely high intensity interactions because *k-means* will produce communities whose union is exhaustive.

## 3 Results

### 3.1 Cumulative clusters

**Fig.(6)** shows the different cumulative clusters, found at time $50 - 58$ in the simulation, in different colors. By this time the double jet has undergone instability and coherent vortices, as well as vorticity filaments, are formed **Fig.(1d)**. As explained earlier, cumulative clusters are formed by particle-particle interactions that occur up to a particular time. The threshold separation $\epsilon$ for interaction between two particles is $40\%$ of the grid spacing in this case. We can see in this figure how different clusters

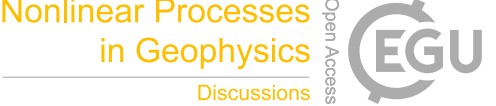

merge during their evolutions. An example for this is the transition from time $52$ to $53$ in **Fig.(6)**, where the green and magenta clusters merge into one magenta cluster. Two clusters merge into one when a particle from one cluster interacts with a particle from another cluster. A question that follows is "Can new clusters take the place of old clusters when they merge?" The answer is yes, we can easily show the forming of new clusters having size of the same order. We create another figure, **Fig(7)**, which

5    is identical to **Fig.(6)**, except for the threshold interaction distance $\epsilon$ set to equal $20\%$ of the initial spatial grid spacing now. Comparing **Fig.(6)** and **Fig.(7)**, we see that the clusters in the later are smaller than those in the first. This is obvious because fewer particles interact with a threshold distance equal to $20\%$ of the grid spacing. In particular, particles in the clusters shown in **Fig.(7)** interact more strongly than those in **Fig.(6)**, and hence the clusters do not evolve the same way in the two cases. Specifically the clusters in the smaller $20\%$ case, do not change size or merge, and their paths are more or less periodic moving

10   around the coherent vortex.

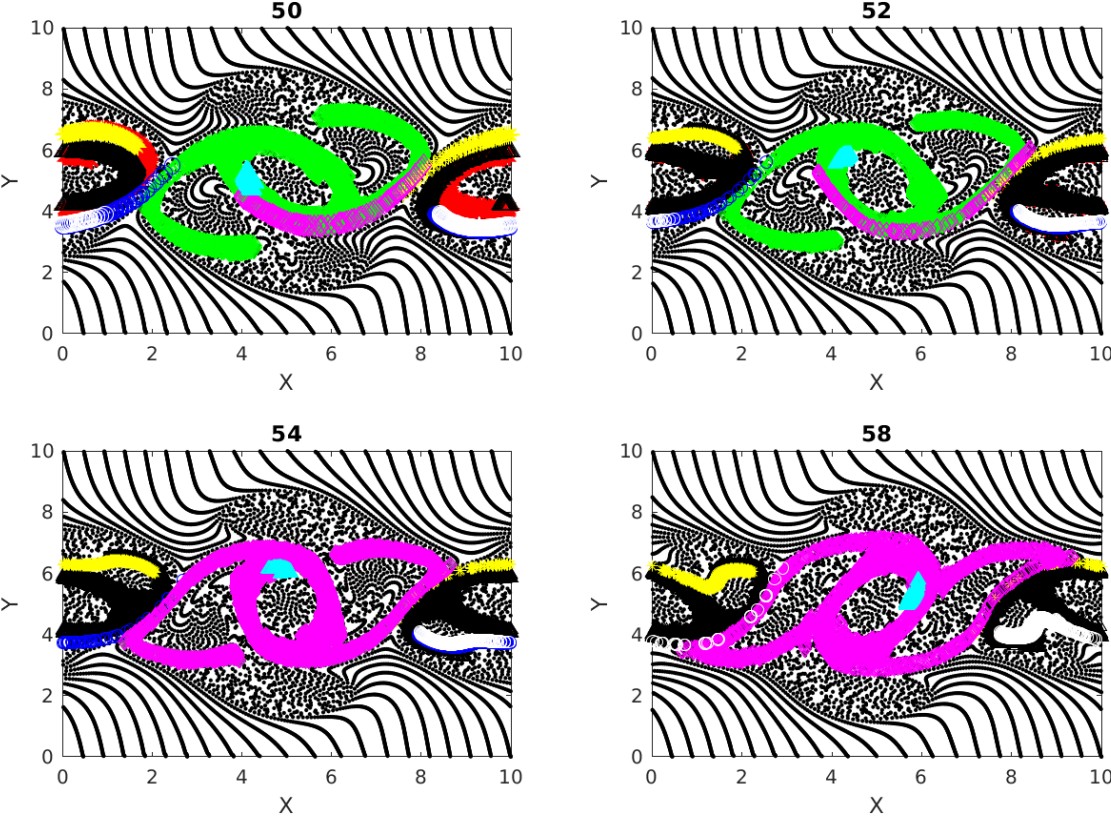

**Figure 6.** Cumulative clusters found at time $50$ with threshold distance for interaction, $\epsilon = 40\%$ of initial separation of particles on uniform rectangular grid and tracked at later time steps. Changing colors notify merging of two clusters when particles from two clusters interact.





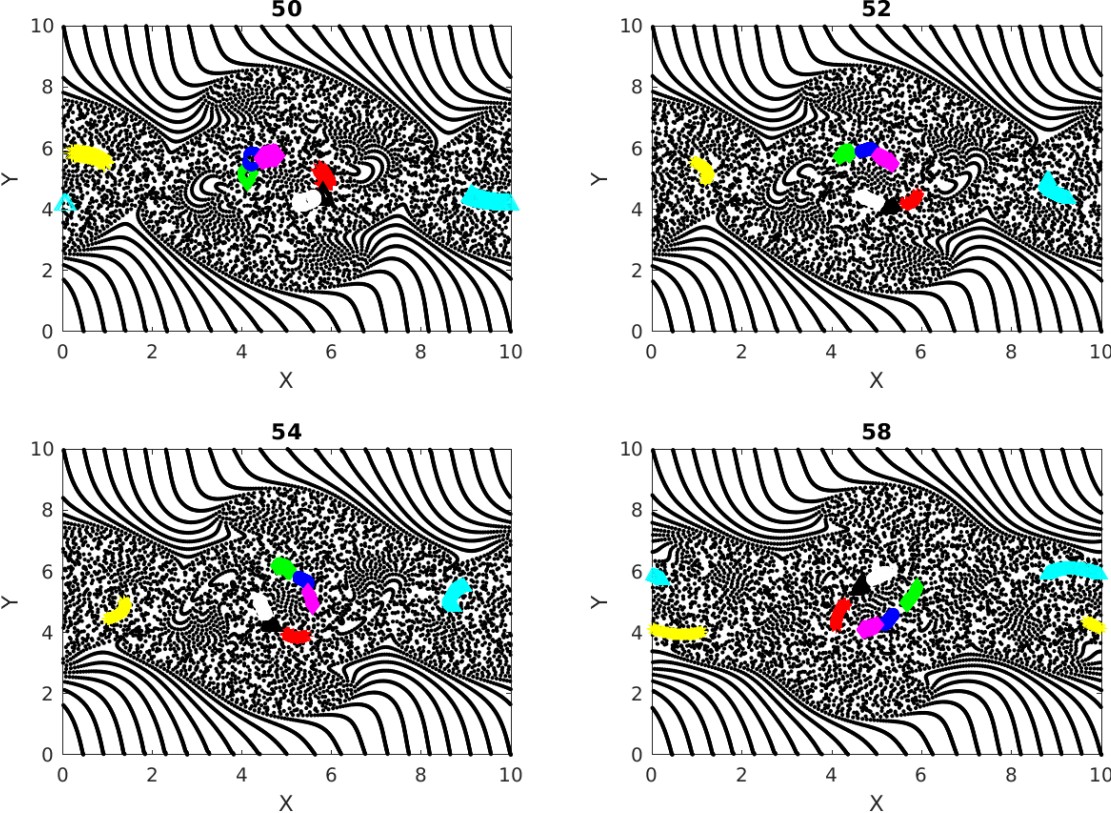

**Figure 7.** Cumulative clusters found at time $50$ with threshold distance for interaction $\epsilon = 20\%$, of initial separation of particles on uniform rectangular grid and tracked at later time steps.

## 3.2 Dense sub-clusters

**Fig(8)** shows the four largest cumulative clusters with $\epsilon = 40\%$ of the grid spacing, found at time $50$ (particles in black) and also plots the dense subclusters mined from within these clusters (particles in blue). We number these clusters as cluster **1**, **2**, **3** and **4** in descending order of their sizes. Recalling the graph theoretic terminology from section 2.3.1, we know each of these subclusters is a graph with a minimum degree of $5$. Dense subclusters locate the regions in a cluster where there are many interactions among particles, significantly more than regions which are not blue. In simpler words these are places where particle interactions are at their peak. Interestingly, the blue regions in this figure have many similarities with the clusters in **Fig(7)**, which represents the stronger interactions. This tells us that the regions of stronger interactions are not very different from the regions of denser interactions in our double-jet flow.





Fig.(9), (10) and (11) show the temporal evolution of cumulative clusters **1, 2** and **3** respectively and the temporal evolution of the particles in the dense-clusters. **Fig.(10)** is different from **Fig.(9)** and **Fig (11)** in the sense that some particles forming the dense sub-clusters in this figure appear to split from other particles in the dense subgroups. This means that particles from these regions of dense interactions move out of their more or less periodic paths and mix with particles in other regions of the flow. In this particular case, it can be said that since these particles undergo dense and also strong interactions they can share physical properties with other particles in the dense clusters, and when they move out of their periodic paths to mix with other particles they interact again and transfer some of their properties to the new regions.

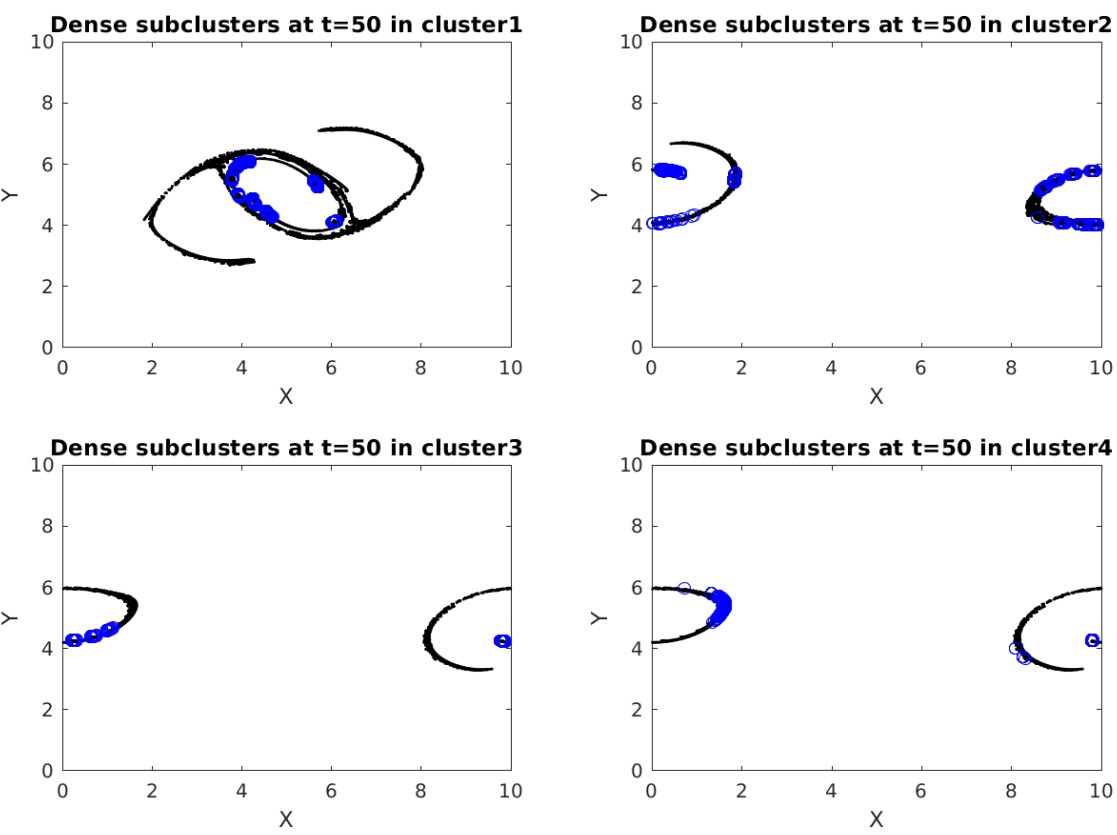

**Figure 8.** Top four (1 being the largest) cumulative clusters (black) with their dense sub clusters (blue) found at time 50. Spatially separated blue regions are distinct sub clusters with each of them having a minimum degree of 5 within themselves and hence called dense.

### 3.3 Instantaneous clusters

**Fig.(12)** shows the temporal behaviour of several of the largest instantaneous clusters found at output time 50. The instantaneous clusters at time 50 seem to be aligned along the boundary of the central vortex, showing that a large group of mutually


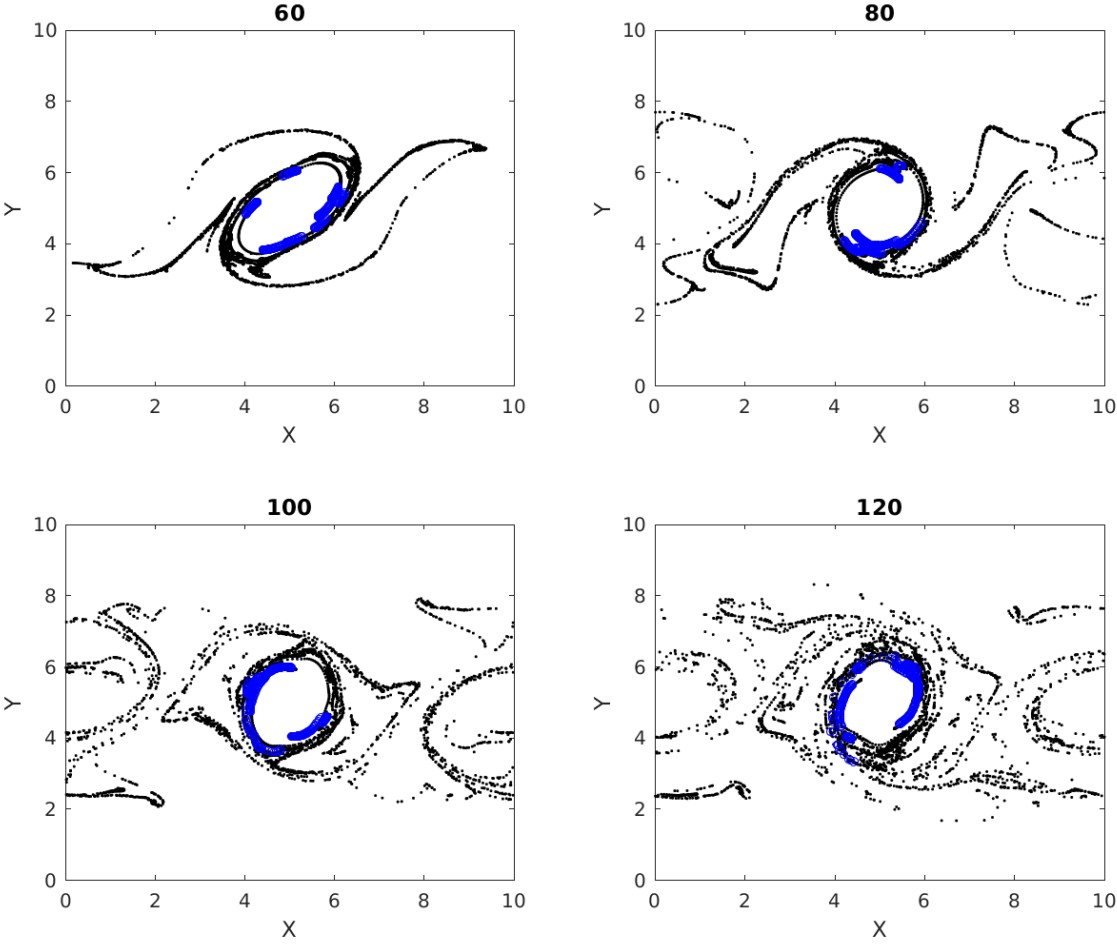

**Figure 9.** Multiple time images of cumulative cluster 1 (black) with its dense sub clusters (blue). Blue 'o's at later times represent particles that were parts of a dense sub cluster at time 50.

interacting particles is concentrated in this region. Instead of finding new clusters at later times, we track the position of these clusters through later times, because this way we check what happens to the highly interactive particles at time 50. It turns out that, these clusters keep moving around inside the central vortex. This means, the already highly interactive particles undergo more interactions with particles in the same region implying higher chances of mixing in the region. Not surprisingly, we draw

5   the same inference from **Fig. 9**, corroborating the authenticity of our algorithm mining regions of dense mixing.



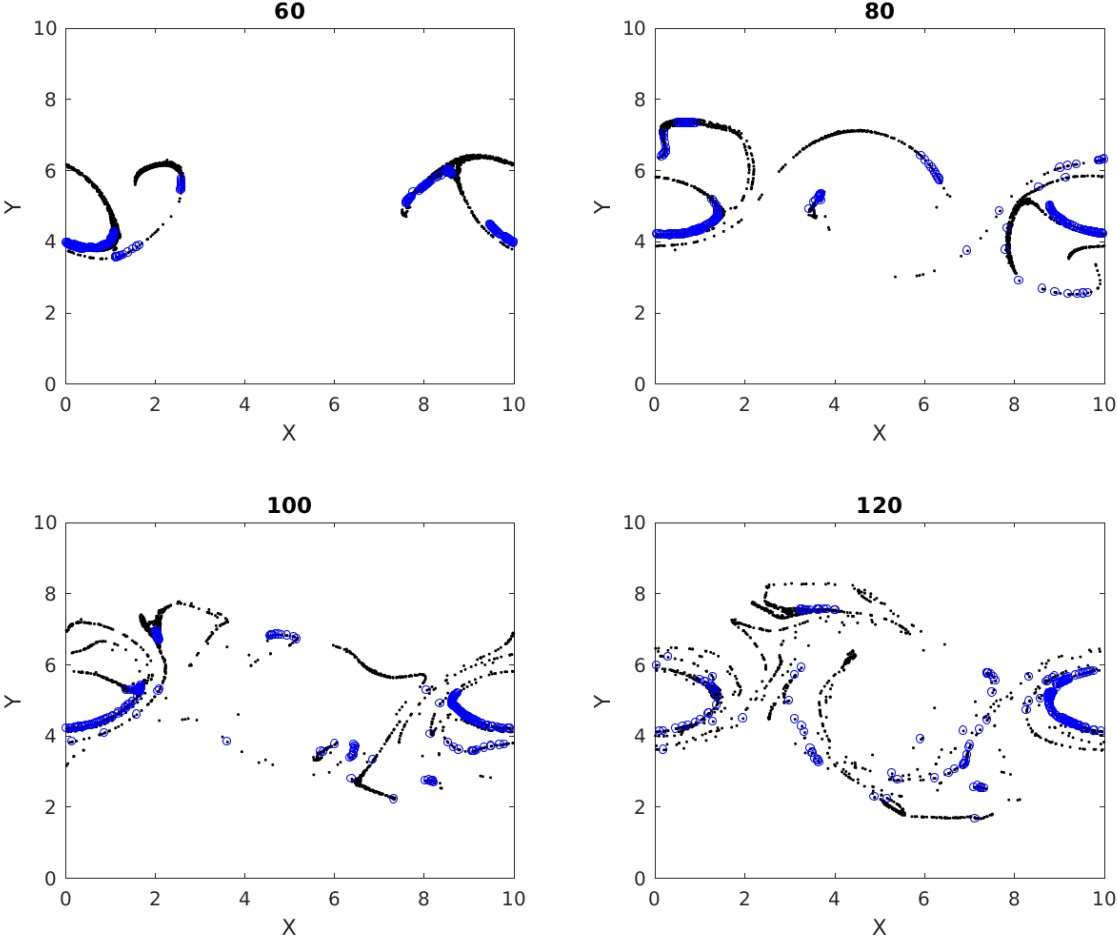

**Figure 10.** Multiple time images of cumulative cluster 2 (black) with its dense sub clusters (blue). Blue 'o's at later times represent particles that were parts of a dense sub cluster at time $50$.

### 3.4 Spectral Clusters

In this sub-section we show the results of spectral clustering described in section 3.4. **Fig.(13)** shows the different spectral sub-clusters that this algorithm splits the largest cumulative cluster (cluster 1) into. **Fig.(14)** shows the evolution of the spectral clusters found at time 50. Giving a quick recap, the spectral clustering technique is responsible for dividing the set of particles into $k$ communities, $k$ being 5 in the results shown. A spectral sub-cluster is expected to have more inter-particle interactions inside itself than outside because the clustering is applied on the adjacency matrix of particle interactions. However, the clusters found here are exhaustive and therefore unlike the dense sub-clusters, all the spectral sub-clusters are not equivalently rich in



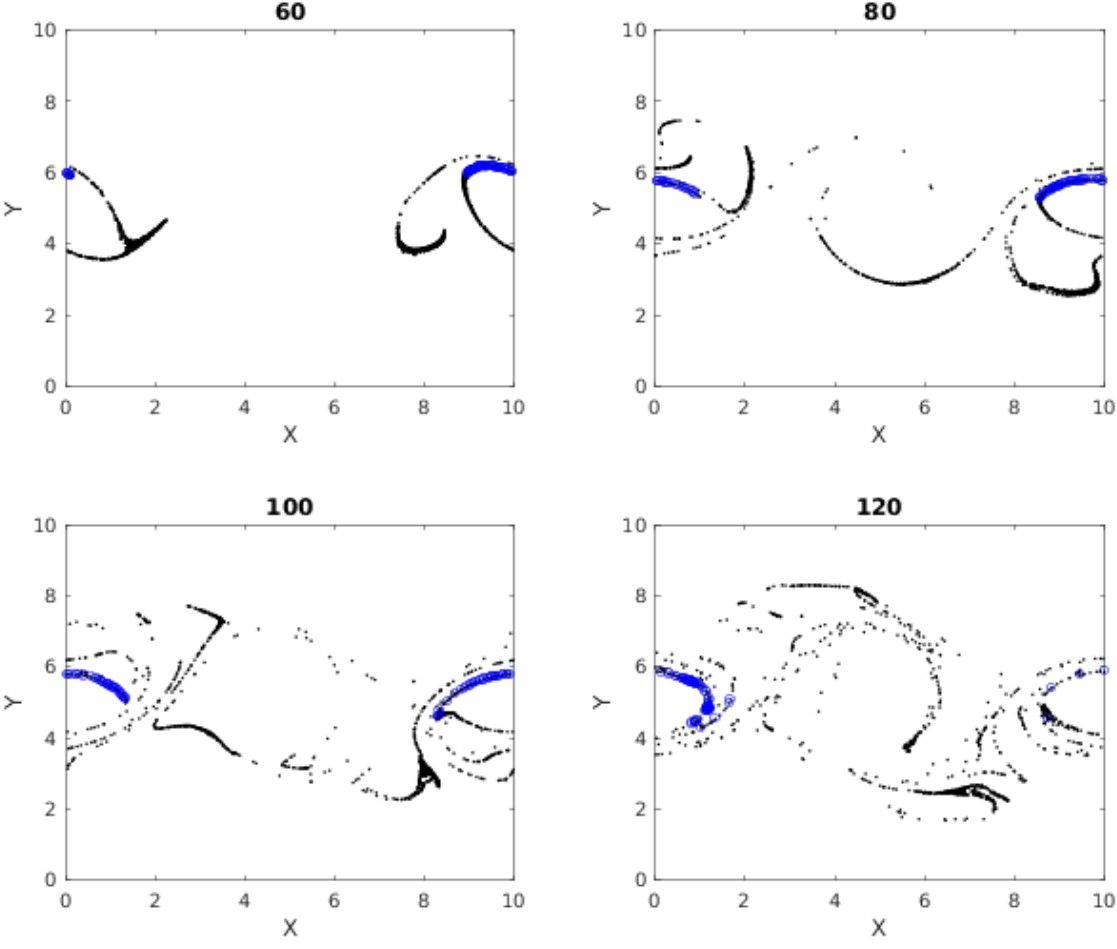

**Figure 11.** Multiple time images of cumulative cluster 3 (black) with its dense sub clusters(blue). Blue 'o's at later times represent particles that were parts of a dense sub cluster at time $50$.

particles with high degrees of interaction. This can be seen from **Fig.(14)** where most of the particles in the sub-clusters of cluster **1** stay within the central vortex, while some others take different paths over the course of the flow's evolution. This can be explained by our hypothesis that the paths of the densely interactive particles in cluster **1** tend to stay nearly periodic with time. Examining **Fig.(13)**, we realize that the spatial distribution of these clusters share similarities to some extent with the dense sub-clusters from the last sub-section, especially around the coherent central vortex. This validates that these coherent structures are home to all the blue regions around the central vortex in **Fig.(9)** representing dense interactions and thereby strong mixing. However, it is clear that the graph theoretic method is more robust in finding specific regions of mixing as compared to the spectral clustering method.





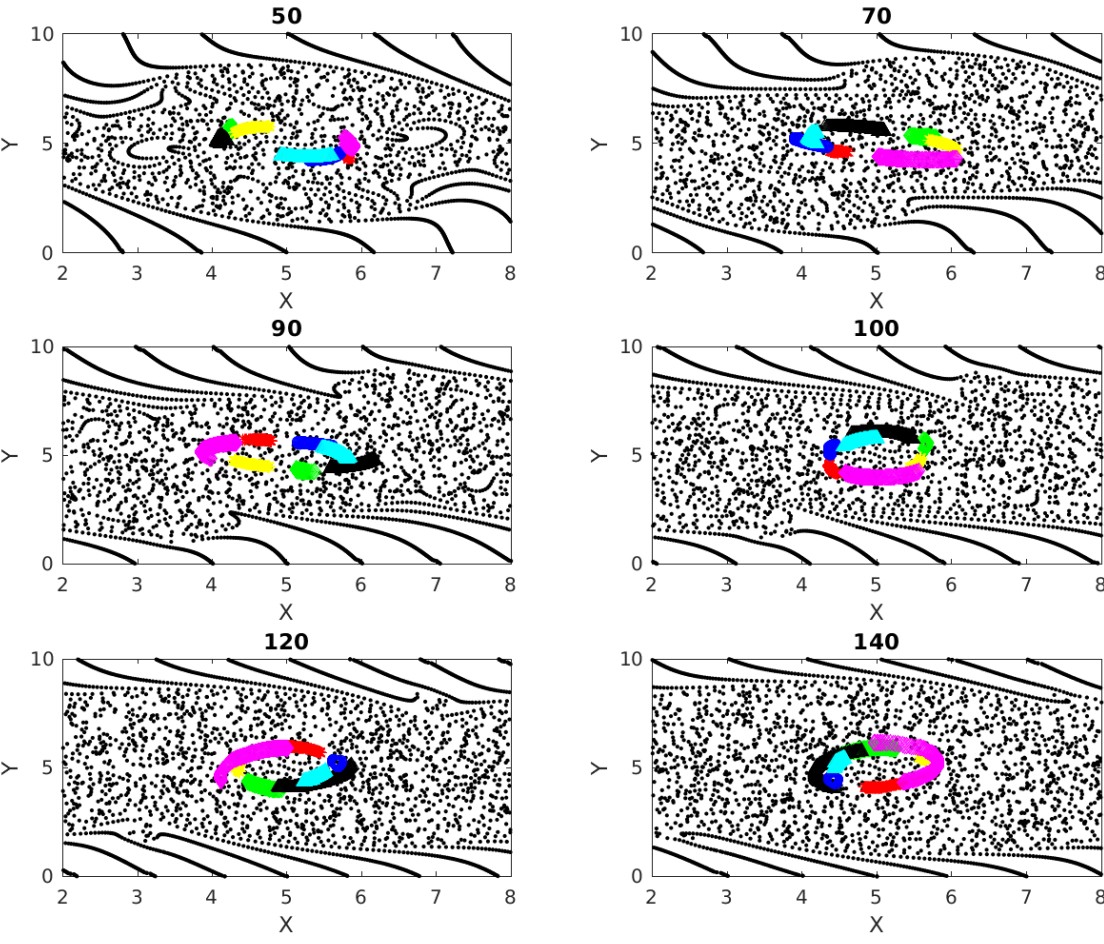

**Figure 12.** Multiple time images of top few instantaneous clusters found at time 50. Once found, particles in these clusters are tracked through later time steps.

We also show the spectral clusters in cluster **2** identified at time 50 in **Fig.(15)** and look at the behaviour of the particles at different times. Comparing with **Fig.(10)**, we see that the particles in the dense clusters show a lot of similarity with the particles in the spectral sub-clusters, especially in the way they deviate from their initial paths and mix into other regions of the flow.



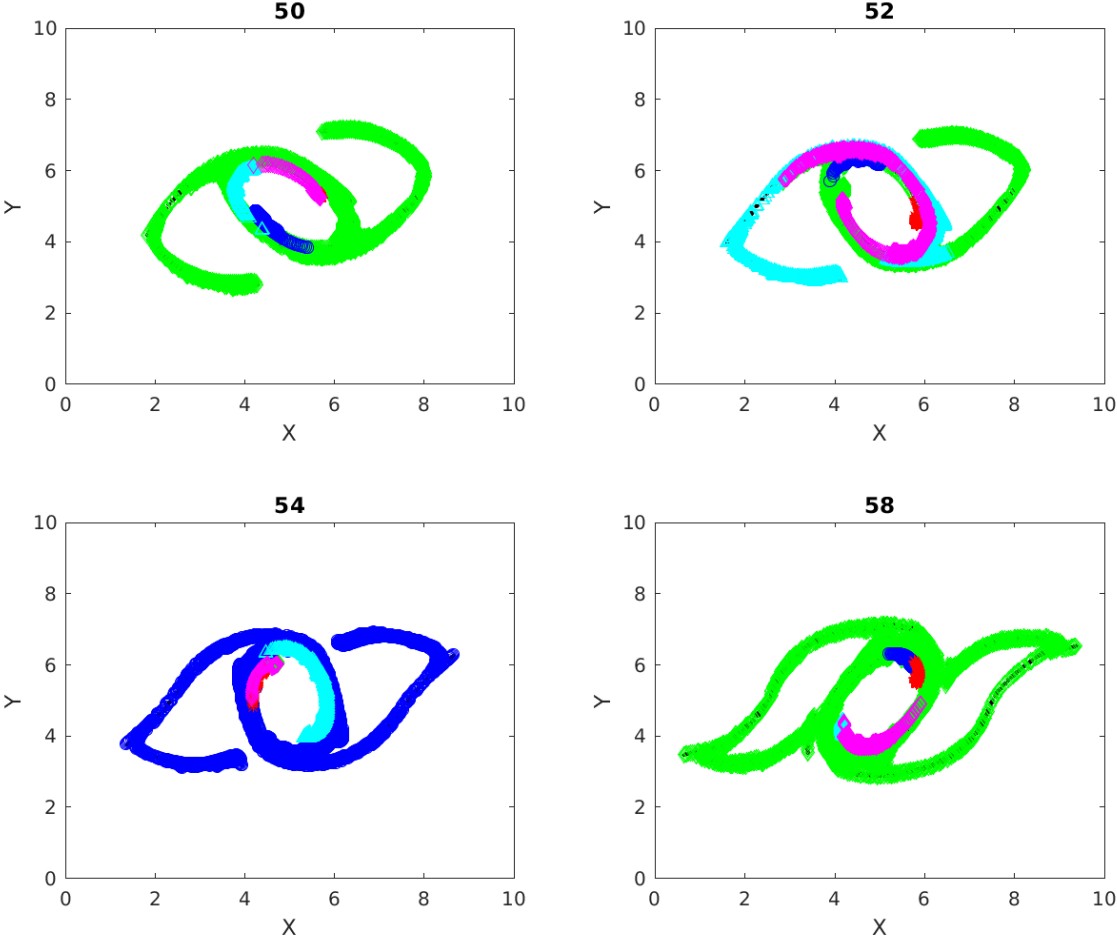

**Figure 13.** Spectral clusters found at multiple times from within cluster 1.

## 4   Conclusions

In this paper we have outlined a Lagrangian-particle based technique to gain insight into mixing in non-linear geophysical flows. Our literature survey showed that clustering of particles based on inter-particle distances has been used to characterize mixing from a Lagrangian point of view. Local network measures like node degree and the local clustering coefficient of a particle, employed by previous researchers e.g. (Padberg-Gehle and Schneide, 2017), gives an idea about the number of other particles a chosen particle has interacted with, or 'neighbours'. We have taken this approach one step further, by finding sub-clusters representing regions of dense interactions. The findings of our work can be partly summarized by **Fig.(16)**. In this figure we examine the output time 80, at which the double jet has broken up into a number of quasi-coherent vortices, as well as





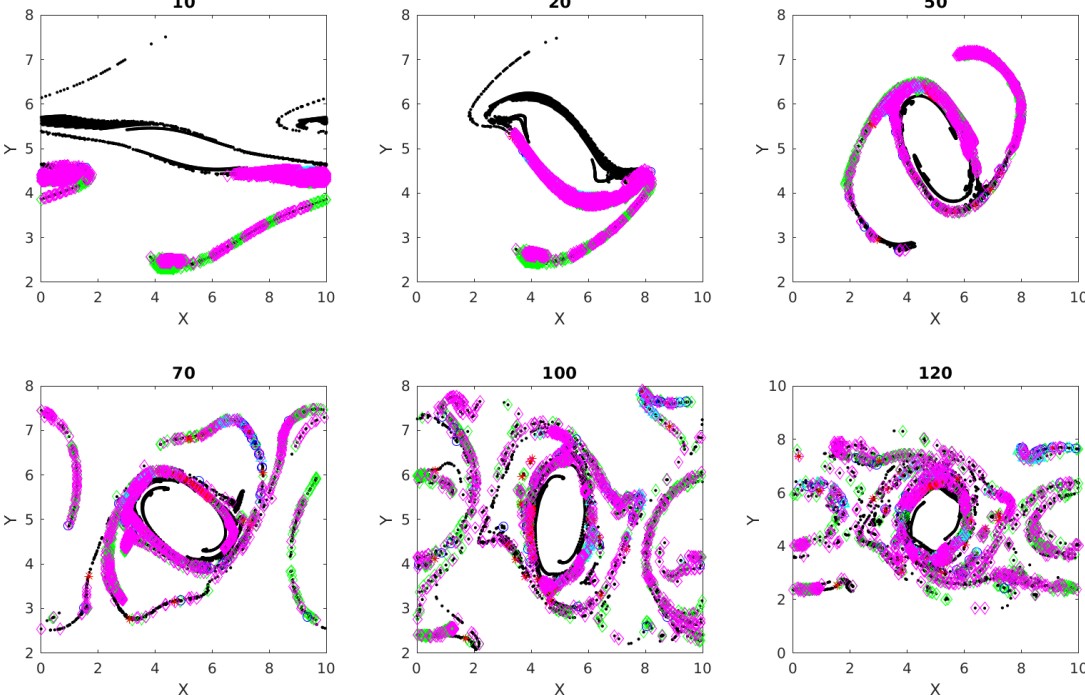

**Figure 14.** Spectral clusters in cluster 1 found at time 50 and tracked forward and backward

filaments of vorticity. The enstrophy field, scaled by its maximum, is shown shaded in the Figure, with green dots superimposed to show particles from a few of the largest cumulative clusters. This gives us an indication of particles that have passed through regions where mixing has taken place. The algorithm *Quick* is used to identify subclusters of particles with dense mutual interactions (i.e. strongest mixing). These particles are plotted in blue. These particles, and their path history, identify regions where the density of mixing is relatively higher (regulated by a density parameter $\gamma$) than other portions of the cumulative clusters. In summary, this figure tells us that the outskirts of the large, coherent vortices involve the most mixing. The vorticity filaments away from the quasi-coherent vortices are marked as belonging to regions of mixing, but not the strongest mixing. The subclustering method thus provides a way to gain further detail on mixing intensity from a Lagrangian point of view.

We have compared our results with the coherent structures identified by spectral clustering. Spectral clustering shows that the location of the coherent structures is around the vortices, but fails to point out the regions of strong mixing. As discussed in section [3.4], the method of finding dense clusters is more precise and robust. We have also computed instantaneous clusters, which as opposed to the cumulative clusters represent regions of interaction for each output time. Instantaneous clusters proved useful in showing that they do not change their paths much during the course of the flow evolution and keep interacting with





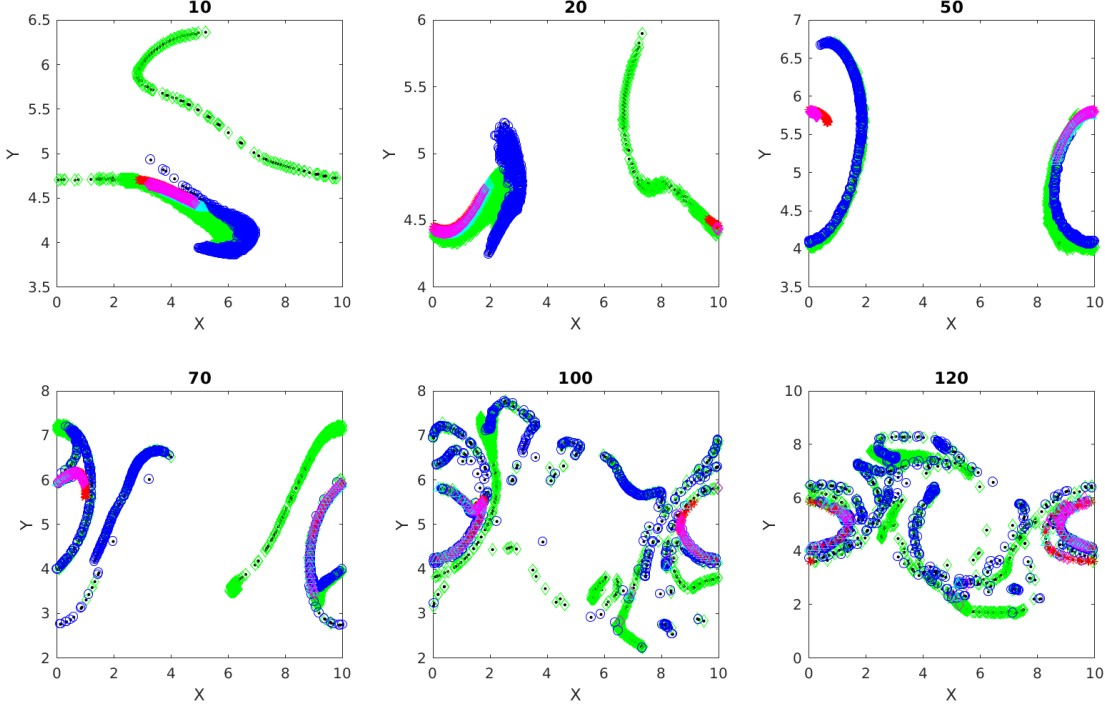

**Figure 15.** Spectral clusters in cluster 2 found at time 50 and shown at other times.

particles in the same region multiple times, implying dense mixing. This helped us validate our method for finding dense subclusters.

Summarizing the major findings in our work, we have seen that the size of cumulative clusters depend on the threshold interaction distance $\epsilon$. In fact previous works like (Padberg-Gehle and Schneide, 2017) have only used values of $\epsilon$ larger than the

5 grid spacing, in order to make the entire graph connected and then apply techniques like spectral clustering to extract coherent sets. Our approach, has allowed us to regulate $\epsilon$ smaller than the grid spacing and observe the differences. We have inferred that, cluster merging is possible beyond a threshold $\epsilon$. Decreasing $\epsilon$ less than the threshold corresponds to stronger interactions and hence stronger mixing. Regions of strong and dense mixing show a lot of similarity, which mostly are concentrated along the outskirts of the quasi-coherent vortices implying that coherent behavior can involve a lot of mixing as demonstrated in

10 **Fig.(16)**. The highly interactive particles from the dense subclusters usually stay as a part of their original coherent vortex. However, interesting dynamics seem to be present when some of these particles deviate out of their usual paths and mix with other regions in the flow as discussed in section [2.3]. Even results from spectral clustering show that some particles showing coherent behaviour may become incoherent over time. The striking similarities between the behaviour of the coherent spectral clusters and the dense subclusters indicate that dense interaction and thereby mixing is a characteristic of coherent structures.

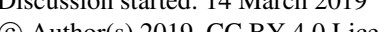



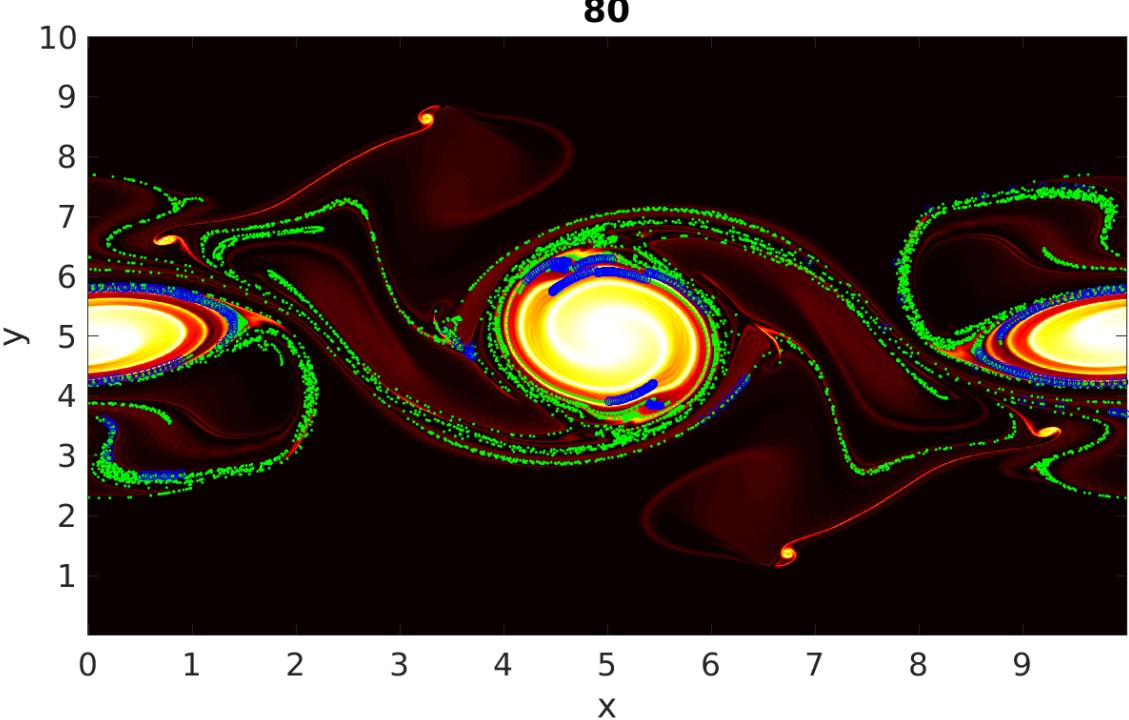

**Figure 16.** Enstrophy field with particles at output time 80. The green dots represent particles from the three largest cumulative clusters and the blue regions represent particles having dense interactions within these cumulative clusters.

Future work divides into algorithmic improvements and applications. On the algorithmic side, we would like to automate the selection of search parameters ($\gamma$ and $min\_size$) in *Quick*, based on the adjacency matrix. A GPU-based implementation of the Shallow Water Equation solver, the Lagrangian particle tracking and dynamic calculation of the inter-particle interactions will also be presented in a future manuscript. On the application side, the central future challenge is how to appropriately think

5  of particles, and hence Lagrangian based mixing ideas, in more complex models. For example should particles migrate across isopycnal layer boundaries in multi-layer models?

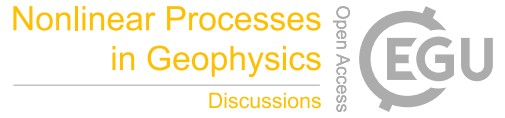

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
