# Peer review of "Particle Clustering and Subclustering as a Proxy for Mixing in Geophysical Flows"

_Nonlinear Processes in Geophysics, 2019_

## Referee Comment (RC1) · Anonymous Referee #1 · 2 May 2019

**Referee report on "Particle Clustering and Subclustering as a Proxy for Mixing in Geophysical Flows" by Rishiraj Chakraborty et al.**

**General comments**

The study of Lagrangian coherent behaviour by means of clustering approaches is a research topic of current interest. In this manuscript, the authors apply a graph-based method to identify regions of dense mixing in a specific example of a two-dimensional flow (shallow water equations on the $f$-plane). The work builds on the trajectory-network approach introduced by Padberg-Gehle and Schneide (2017), where a network is built with Lagrangian trajectories serving as nodes. Two nodes are linked when the underlying trajectories have come $\varepsilon$-close at least once in the time span under consideration, which is coded in a binary adjacency matrix. While in Padberg-Gehle and Schneide (2017) $\varepsilon$ is chosen in dependence of the initial grid size and the particle density in order to ensure a connected but sparse network, the authors of the manuscript under review concentrate on local, dense mixing regions. For this they consider the cut-off radius $\varepsilon$ to be smaller than the mesh size of the grid where particles are initialized. The resulting network is then typically disconnected. The largest connected components, which the authors call clusters, are then further studied using two community detection approaches, namely the *Quick* algorithm introduced by Liu and Wong (2008) to find maximal quasi-cliques (i.e. dense subgraphs) and the classical spectral method by Shi and Malik (2000) for the solution of a normalized cut problem. The methods are tested on one specific example system.

While the manuscript is well written, contains some original ideas and altogether adds to the current research in the cluster-based study of Lagrangian coherence, there are some issues that have to be worked on in my view before the manuscript can meet the high standards for publication in NPG. Major issues include the question of the robustness of the methods, the lack of a critical comparison with other (network-based) mixing indicators, as well as the question of the relevance of the detected regions.

**Specific comments**

1) The detection of dense sub-clusters by means of the *Quick* algorithm is stressed several times throughout the manuscript as giving more robust results compared to spectral clustering. This however is not convincingly demonstrated in the paper. In particular, it would be interesting to see how the detected dense subclusters depend on the cut-off radius $\varepsilon$ in the network construction and on the two quasi-clique parameters. While the cumulative clusters already differ significantly for two different values of $\varepsilon$ (40% and 20% mesh size, Fig. 6/7), I assume that the dense sub-clusters as shown in Figs. 8--11 would vary considerably depending on the three parameters in the method. If a very small $\varepsilon$ is chosen, then I also expect that the results would differ when the initial grid points are shifted. Also, how do the dense subclusters look like when $\varepsilon$ is chosen larger than the mesh size? The results of corresponding numerical studies (also for instantaneous clusters) should be presented in the paper.

2) Adding to 1): When comparing Fig. 6 and 7 (p13) can any conclusions about the "perfect" thresholding distance $\varepsilon$ be made?

3) The relation to other graph properties (e.g. as addressed on p20, l4) is not explored at all. For instance, can the detected structures also be related to a large node degree and/or a large local clustering coefficient? The manuscript would greatly benefit from a corresponding numerical comparison.

4) On p22 (l13) the authors write that "The striking similarities... indicate that dense interaction and thereby mixing is a characteristic of coherent structures." The dense subclusters appear to be located at the boundary of coherent vortices, but do not make up the entire boundary, which however may

be specific to the choice of parameters and initial conditions. The overall relevance of the detected small structures for transport and mixing remains unclear to me as mixing here seems to be very localized. Also, depending on the choice of parameters, the detected regions and their interpretations may differ significantly (see also point (1) above).

5) On p22 (l1) the authors write: "This helped us validate our method for finding dense subclusters." This statement refers to a comparison of cumulative clusters plus subclustering and instantaneous clusters. If both approaches find the same regions here then it would be interesting for the reader which way is less expensive and which way is more robust.

6) The clustering approach proposed in the manuscript has also some relation to the concept of the trajectory encounter volume as introduced by Rypina, Pratt (NPG, 2017). The authors should refer to this work as well.

7) Section 2.3.1: The description of the Quick algorithm by Liu and Wong (2008) is very technical. As the details are not referred to later in the text, the authors should focus on the main idea of the algorithm and delegate the details to an appendix.

8) In Figure 5, I assume that with the given parameters, the clique of size 4 could be extended by including the node right next to this subgraph (?).

9) Fig. 2/3: In general, an adjacency matrix only has only 1s on the diagonal in the case of self-loops, which is not the case in this construction.

10) Figs. 2--4 can be merged into one.

11) In the introduction many different methods for studying Lagrangian coherence are discussed, but corresponding references are missing.

12) p12 (l18): A reference to Shi & Malik (2000) for the normalized cut problem is missing.

13) p13 Fig. 6: A "transition from time 52 to 53 in Fig. 6" is mentioned in the text, but there is no time frame 53 in Fig. 6. In the caption it says: "Cumulative clusters … tracked at later times". However, this would show the particles coloured according to first time step but plotted at later times. From the idea of cluster merging etc. I assume the clustering is performed individually for each of the plotted times.

14) In view of including further numerical studies, the authors should consider condensing the presentation of some the current results that are demonstrated in very much detail in Figs. 6-16.

**Technical comments (typos, etc.)**

- p11 l18 & 23: missing {
- p.3 l. 4: Hadjighasem et al …. "However, these principles only apply in the early stages…" should rather be something like: "only apply in finite time intervals…"
- p.9 l. 1: "We find sub-clusters with a minimum size of…" rather say "We search for sub-clusters with a minimum size of … throughout our analysis of the double jet flow…".
- p17 l3 should rather be "Fig. 14 shows the temporal evolution of the spectral sub-clusters of cluster 1 found at time 50."
- p21 l5 "…identify regions where the density of mixing is relatively higher than other portions of the cumulative clusters." What is the "density of mixing"?
- p21 l6 "… involve the most mixing." Rather say "strongest mixing."

---

## Referee Comment (RC2) · Anonymous Referee #2 · 9 May 2019

The manuscript "Particle Clustering and Subclustering as a Proxy for Mixing in Geophysical Flows" by Chakraborty, Coutino, and Stastna addresses the problem of mixing in geophysical flows by means of a particle-based Lagrangian approach. The authors identify clusters and subclusters of particles in their simulation and draw conclusions on the flow based on them.

What I like in the paper is the careful description of the theoretical background, which takes a large part in the text but it is definitely important to understand the results.

There are two main points that I see problematic in the paper:

- In a paper based on simulations I would expect some critical discussion about the influence of the numerics on the results. In the manuscript this is missing, although in

principle the topic of mixing cannot be treated without considering what happens near the resolution scales. I would ask the authors to add details about it, like for instance a resolution study or more in-depth considerations on the numerical tools that they are using, and how they can affect their results.

- Despite of the detailed theoretical description, most of the analysis of the results is based on a qualitative assessment of the figures. Would it be possible to define some quantitative diagnostics to support what the authors infer?

Minor points:

- the style of citations should be improved. Not everything should go in brackets, i.e. sometimes \citet should be used instead of \citep (assuming the authors used LaTeX for editing);

- p.8, eq. (4): do I understand correctly that gamma is in the interval between 0 and 1? If it is the case, please mention in the text.

Besides these comments, I think that the paper meets the quality and scientific standards for publication on NPG. I would recommend to accept it, after the points listed above are properly taken into account by the authors.

---

## Author Comment (AC1) · 24 Jun 2019

We thank the reviewer for their comments, and we have modified the manuscript substantially based on the the reviewer's suggestions. We provide detailed discussion in bold below each of the reviewer's comments and the changes in the manuscript are denoted in red.

**Comment 1**

1)Referee comment - The detection of dense sub-clusters by means of the Quick algorithm is stressed several times throughout the manuscript as giving more robust results

compared to spectral clustering. This however is not convincingly demonstrated in the paper. In particular, it would be interesting to see how the detected dense subclusters depend on the cut-off radius e in the network construction and on the two quasi-clique parameters. While the cumulative clusters already differ significantly for two different values of e ($40\%$ and $20\%$ mesh size, Fig. 6/7), I assume that the dense sub-clusters as shown in Figs. 8–11 would vary considerably depending on the three parameters in the method. If a very small e is chosen, then I also expect that the results would differ when the initial grid points are shifted. Also, how do the dense subclusters look like when e is chosen larger than the mesh size? The results of corresponding numerical studies (also for instantaneous clusters) should be presented in the paper.

2) **Response - The purpose of this paper is to extract structures with higher density of interactions reflecting regions of strong mixing. Spectral clustering as introduced in previous literature may fail to be consistent when applied at different output times, because of the clustering algorithms used. It also returns clusters of incomparable sizes, which leaves us no way to compare the degree of mixing among the clusters mined. Our method on the other hand controls the density of connections and hence all clusters mined belong to the same class. We have carried out a study on the effects of varying $\epsilon$ on the cluster size. Increasing $\epsilon$ relaxes the threshold criteria for particle interaction. Thus, at a certain time more particles will be part of a cumulative cluster with increased $\epsilon$. Let's say cumulative clusters with $\epsilon = 40\%$ and $\epsilon = 60\%$ be $C_{40}$ and $C_{60}$ respectively at a time $t = 50$. Since the number of particles in the simulation is constant, $C_{40} \subset C_{60}$ and we have verified this. We found $|C_{60}| \sim 12|C_{40}|$. The time complexity of 'Quick' scales exponentially with increase in size of cluster, average degree and negative $\gamma$, because it is an unsupervised learning algorithm with no a priori estimate. If we focus on $C_{40}$ in $C_{60}$, the average degree of $C_{60}[C_{40}]$ naturally increases. Now, if we want to mine dense clusters from $C_{60}[C_{40}]$, the minimum degree we set has to be more than that we set for $C_{40}$. Hence, the particles in dense clusters mined from $G(C_{40})$ will be a subset of the those mined from $C_{60}[C_{40}]$.**

**Shifting the particles is a good idea as well. The sensitivity analysis of the dense clusters to initial particle position perturbation has been added in the revised manuscript with the same $\epsilon$ as the base case ($40\%$), because $\epsilon$ lower than that doesn't yield any comparable results anyways.**

**As an overall point, the idea of an $\epsilon$ value that is small but not too small is a fairly common argument in continuum mechanics. Our point is not to argue for a particular value and in an application driven setting (e.g. oil spill dispersal) the value would have to be chosen on a case by case basis.**

3)Author's changes in manuscript - We have added a separate sub-section in the manuscript describing the characteristics of the dense clusters. We have added one figure showing dense clusters for $\epsilon = 60\%$ and varying $\gamma$, one figure just showing the effects of varying $\gamma$ on our base case $\epsilon = 40\%$, and figures showing the effect of perturbation of the initial position of the particles. To understand the complete significance of the figures, the corresponding parts of the revised manuscript needs to be read.

**Comment 2**

1) Referee comment - Adding to 1): When comparing Fig. 6 and 7 (p13) can any conclusions about the "perfect" thresholding distance $\epsilon$ be made?

2) **Response- Theoretically, the lower the value of $\epsilon$ which can give us an understanding about the dense clusters, the better. However, since a spatial discretization is used a practical lower bound (below which the numerical method cannot provide information) must exist. For example, in our case $\epsilon = 20\%$ is too small to mine subclusters with a meaningful minimum degree. Therefore we must take $\epsilon > 20\%$. But as soon as we find a satisfactory number of sub-clusters with density more than other regions, increasing $\epsilon$ is always guaranteed to include the already identified regions. We realize this at $\epsilon = 40\%$. For practical purposes,**

**it is actually necessary to find the $\epsilon$ and minimum degree which works for the problem and provides some meaningful insight. Increasing $\epsilon$ more than necessary increases the degree of vertices thereby increasing the time needed for the computation exponentially. We agree that this introduces some subjectivity into our methodology but at least this is done in a transparent way.**

3)Author's changes in manuscript - The above argument has been included in the manuscript, new simulations supporting the argument have been added in the revised manuscript and figures added mentioned in changes against comment 1.

**Comment 3**

1) Referee comment - The relation to other graph properties (e.g. as addressed on p20, l4) is not explored at all. For instance, can the detected structures also be related to a large node degree and/or a large local clustering coefficient? The manuscript would greatly benefit from a corresponding numerical comparison.

2) **Response- We have provided a comparison to local clustering coefficients and node degree in the revised manuscript.**

3)Author's changes in manuscript - We have added a figure and provided comparison to local clustering coefficients and node degree in the revised manuscript for the top 4 cumulative clusters at output time $50$.

**Comment 4**

1) Referee comment - On p22 (l13) the authors write that "The striking similarities... indicate that dense interaction and thereby mixing is a characteristic of coherent structures." The dense subclusters appear to be located at the boundary of coherent vortices, but do not make up the entire boundary, which however may be specific to the choice of parameters and initial conditions. The overall relevance of the detected small structures for transport and mixing remains unclear to me as mixing here seems to be very localized. Also, depending on the choice of parameters, the detected regions and their interpretations may differ significantly (see also point (1) above).

2) **Response- We have discussed choosing $\epsilon$ above. The minimum degree is controlled by minimum size and $\gamma$. The greater the minimum degree, the better the clusters represent localized mixing. Thus we choose as high a minimum degree as we can, i.e. which gives us a satisfactory number of clusters in a satisfactory amount of time. We are proposing to mimic localized mixing by particle interactions (following existing literature). A dense sub-cluster has more particles interacting among each other, so more localized mixing might be taking place. As we noted above our methodology does not remove all subjectivity from the problem, but the subjectivity present at least has a logical reason for requiring the user to make a choice of 'best' $\epsilon$.**

3) Author's changes in manuscript - Already mentioned against comment 1.

**Comment 5**

1) Referee comment - On p22 (l1) the authors write: "This helped us validate our method for finding dense subclusters." This statement refers to a comparison of cumulative clusters plus sub clustering and instantaneous clusters. If both approaches find the same regions here then it would be interesting for the reader which way is less expensive and which way is more robust.

2) **Response- For our specific example, the biggest instantaneous clusters are always found near the boundary of the central vortex, which just acts as a partial check on whether our dense clusters makes sense. Naturally mining the instan-**

**taneous clusters is much cheaper than the dense quasi cliques.**

3)Author's changes in manuscript - Since the instantaneous clusters don't contribute much to the key idea of our work and in order to remove ambiguity the authors decide to take it off the manuscript.

**Comment 6**

1) Referee comment - The clustering approach proposed in the manuscript has also some relation to the concept of the trajectory encounter volume as introduced by Rypina, Pratt (NPG, 2017). The authors should refer to this work as well.
2)  **Response- this paper has been discussed in the revised text.**

**Comment 7**

1) Referee comment - Section 2.3.1: The description of the Quick algorithm by Liu and Wong (2008) is very technical.  As the details are not referred to later in the text, the authors should focus on the main idea of the algorithm and delegate the details to an appendix.
2)  **Response- The technical description does not strike the authors as too long and based on the second reviewer's comments, it appears to be appreciated. Thus we have decided to keep it in its original location.**

3)Author's changes in manuscript -No change.

**Comment 8**

1) Referee comment - In Figure 5, I assume that with the given parameters, the clique of size 4 could be extended by including the node right next to this subgraph (?).
2) **Response- We have increased $\gamma$ from $0.3$ to $0.4$, to avoid the anomaly.**

**Comment 9**

1)Referee comment - Fig. 2/3: In general, an adjacency matrix only has only 1s on the diagonal in the case of self-loops, which is not the case in this construction.
2) **Response- The principal diagonal in the adjacency matrix illustration has been replaced with $0$s in the revised text.**

3)Author's changes in manuscript - We have added new illustrations with the above changes.

**Comment 10**

1)Referee Comment - Figs. 2–4 can be merged into one.
2) **Response- The corresponding change has been incorporated into the revised manuscript.**

**Comment 11**

1)Referee Comment - In the introduction many different methods for studying Lagrangian coherence are discussed, but corresponding references are missing.

2) **Response- The references for the methods probabilistic transfer operator, dynamic Laplace operator and the hierarchical coherent pairs have been added in the revised manuscript.**

**Comment 12**

1)Referee Comment - p12 (l18): A reference to Shi  Malik (2000) for the normalized cut problem is missing.

2) **Response- Corresponding reference has been added in the revised text.**

**Comment 13**

1)Referee Comment - p13 Fig. 6: A "transition from time 52 to 53 in Fig. 6" is mentioned in the text, but there is no time frame 53 in Fig. 6. In the caption it says: "Cumulative clusters . . . tracked at later times". However, this would show the particles coloured according to first time step but plotted at later times. From the idea of cluster merging etc. I assume the clustering is performed individually for each of the plotted times.

2) **Response- We are tracking the evolution of the clusters identified at time $50$.**

3)Author's changes in manuscript - The caption in the corresponding figure has been made clearer in the revised text.

**Comment 14**

1)Referee Comment - In view of including further numerical studies, the authors should consider condensing the presentation of some the current results that are demonstrated in very much detail in Figs. 6-16.

2) **Response- We removed the instantaneous clusters section and the second cluster evolution for the spectral clusters i.e. Fig. 12 and Fig. 15 respectively in the old manuscript.**

**Technical comments (typos, etc.)**

1)Referee Comment - p11 l18  23: missing {
• p.3 l. 4: Hadjighasem et al . . .. "However, these principles only apply in the early stages. . ."
should rather be something like: "only apply in finite time intervals. . ."
• p.9 l. 1: "We find sub-clusters with a minimum size of. . ." rather say "We search for subclusters with a minimum size of . . . throughout our analysis of the double jet flow. . .".
• p17 l3 should rather be "Fig.  14 shows the temporal evolution of the spectral sub-clusters of cluster 1 found at time 50."
• p21 l5 ". . .identify regions where the density of mixing is relatively higher than other portions of the cumulative clusters." What is the "density of mixing"?
• p21 l6 ". . . involve the most mixing." Rather say "strongest mixing."

2) **Response- All the above comments have been taken care of in the revised manuscript.**

[Figure]

[Figure]

**Fig. 1.** Dense clusters with $\epsilon=60\%$ in cumulative clusters 1 and 2 at $t=50$.

[Figure]

**Fig. 2.** Dense clusters with $\epsilon=40\%$ for varying $\gamma$ at $t=50$

[Figure]

**Local Clustering Coefficient**

**Node degree**

**Fig. 3.** Local clustering coefficient (top panel) and node degree (bottom panel) for the top four cumulative clusters at output time $50$

**Fig. 4.** Dense clusters with $\epsilon=40\%$ and particles on uniform rectangular grid.

[Figure]

Interactive
comment

[Figure]

**Fig. 5.** Dense clusters with $\epsilon=40\%$ and particles on rectangular grid with perturbations.

---

## Author Comment (AC2) · 24 Jun 2019

We thank the reviewer for their comments, and we have modified the manuscript based on the the reviewer's suggestions. We provide detailed discussion in bold below the each of the reviewer's comments and in the manuscript the changes are denoted in red.

**Comments**

1. In a paper based on simulations I would expect some critical discussion about the influence of the numerics on the results. In the manuscript this is missing,

although in principle the topic of mixing cannot be treated without considering what happens near the resolution scales. I would ask the authors to add details about it, like for instance a resolution study or more in-depth considerations on the numerical tools that they are using, and how they can affect their results.

**Response: A paragraph of discussion on numerics has been added to the text. The spectral method used is close to optimal, for a fixed grid, and along with the grid resolution tests we have carried out, this gives us considerable confidence in the code. The more challenging issue, going forward will be to consider 3D simulations.**

2. Despite of the detailed theoretical description, most of the analysis of the results is based on a qualitative assessment of the figures. Would it be possible to define some quantitative diagnostics to support what the authors infer?

**Response - A quantitative figure regarding the position of the dense clusters has been added to the manuscript. Moreover, theoretical description provided, is about the methods of community detection from a graph. We use this technique to draw inference about characteristics of mixing from a graph.**

3. the style of citations should be improved. Not everything should go in brackets, i.e. sometimes
citet should be used instead of
citep (assuming the authors used LaTeX for editing).

**Response - Appropriate changes have been made in the text.**

4. p.8, eq. (4): 1) do I understand correctly that gamma is in the interval between 0 and 1? If it is the case, please mention in the text.

**Response - It has been mentioned in the revised text.**

[Figure]

**Fig. 1.** Displacement averaged over particles in dense clusters from clusters $1,2,3$ (DC 1, DC 2, DC 3) measured from positions at output time $50$ vs output time.